# Development of an epilepsy self-management mobile health app framework: Content validity study results

**Mohsen Zaied Alzamanan**[1], **Kheng-Seang Lim**[2]*, **Maizatul Akmar Ismail**[3], **Norjihan Abdul Ghani**[3]

**1** Department of Information Systems, University of Malaya, Kuala Lumpur, Malaysia, **2** Division of Neurology, Department of Medicine, Faculty of Medicine, University of Malaya, Kuala Lumpur, Malaysia, **3** Department of Information Systems, Faculty of Computer Science and Information Technology, University of Malaya, Kuala Lumpur, Malaysia

* kslimum@gmail.com

## Abstract

### Background

Mobile health (mHealth) applications (apps) show promise in supporting epilepsy self-management (eSM). To delve deeper into this potential, we conducted a systematic review of epilepsy mHealth apps available on both iOS and Android platforms, examining articles related to eSM. This review allowed us to identify important domains related to eSM. Furthermore, based on the findings, we developed an epilepsy mHealth app framework that aims to improve self-management for the local population. This study aims to assess the practicality and usability of the proposed mHealth app framework designed to improve eSM. We will conduct an expert panel review to evaluate the effectiveness and feasibility of the framework.

### Material and methods

Content validity was assessed by an expert panel comprising epileptologists and pharmacists. The validation process involved scoring the items within each domain of the framework to evaluate their practicality and usability (quantitative component). In addition, a panel discussion was conducted to further explore and discuss the qualitative aspects of the items.

### Results

A total of 4 domains with 15 items were highly rated for their practicality and usefulness in eSM.

### Conclusions

The locally validated framework will be useful for developing eSM mobile apps. Seizure Tracking, Medication Adherence, Treatment Management, and Healthcare Communication emerged as the most crucial domains for enhancing eSM.

**Data Availability Statement:** All relevant data are within the manuscript and its Supporting information files.

**Funding:** The author(s) received no specific funding for this work.

**Competing interests:** The authors have declared that no competing interests exist.

**Abbreviations:** app, application; COVID-19, corona virus disease 2019; eSM, epilepsy self-management; HCP, health care professional; I-CVI, Items Content Validation Index; mHealth, mobile health; WHO, World Health Organization.

## Introduction

Epilepsy is a prevalent neurological condition characterized by unpredictable occurrences [1]. Globally, it affects approximately 50 million people [2], with 1% of the Malaysian population living with epilepsy [3]. A recent study revealed that the mortality rate among individuals with epilepsy is higher than that of the general population, and the lifetime prevalence of epilepsy is 7.8 per 1000 people [4].

Patient with epilepsy (PWE) must actively manage their condition to comprehend their situation, evaluate medication efficacy, and regulate seizures.

According to the World Health Organization (WHO), chronic disorders such as epilepsy pose the most significant challenge to modern healthcare systems [5]. However, the situation varies across countries and raises concerns. For instance, nearly half of the hospitals in the United States have fewer than 100 available beds, with a lack of easily accessible neurologists [6]. This shortage is prevalent even in rural areas where higher cases have been reported [7]. As a result, self-management becomes a top priority for patients with epilepsy and their health care professional (HCP) in order to manage the condition effectively and enhance their quality of life.

Self-management refers to individuals' capability to handle their symptoms, medications, physical and psychosocial effects, and necessary lifestyle changes to effectively cope with a chronic condition, such as epilepsy or cancer [8]. This concept highlights patients' capacity to manage their chronic illnesses with the ultimate objective of optimizing their overall quality of life [9].

Patients with epilepsy and their caregivers face challenges in managing the condition effectively due to the lack of an appropriate tool [10]. Moreover, individuals with epilepsy may experience comorbidities like depression, anxiety, or sleep disorders, making it difficult to recognize symptoms without proper monitoring by HCPs [11]. As a result, epilepsy self-management (eSM) presents a significant challenge for patients with epilepsy and their HCP.

To address this challenge, some trials recommend that patients use a mobile health (mHealth) application (app) to report their health status and symptoms, thereby enhancing self-management. By doing so, the app aids in epilepsy management and supports HCPs in making informed decisions.

mHealth is capable of supporting and assisting health care providers in education, aiding in diagnosis, and facilitating patient management. In addition, it can streamline communication between health care services and patients, making information sharing more efficient and convenient [12].

The term "mHealth app" refers to the utilization of a mobile phone's portability to aid in training, monitoring, self-care, diagnosis, and treatment of diseases [13].

mHealth provides an opportunity to improve patient health and reduce the necessity for office visits in the routine treatment of prevalent acute and long-term diseases. It offers the potential to enhance patient outcomes and make health care more accessible [14].

mHealth enables real-time health assessment at both population and individual levels. It encourages healthy habits to prevent or minimize health issues, supports self-management of chronic illnesses, enhances HCP expertise, and ultimately reduces the frequency of health care visits [15].

Self-management through mobile technology is an effective approach to improving health care services [16].

mHealth apps have shown great promise and have been successful in various fields, particularly in health care. These apps also facilitate patient progress monitoring and management [17].

We initially performed a literature review (published separately [18]) and identified existing features of mHealth apps related to eSM that are available on the app stores (i.e., iOS and Android). Additionally, another literature review was conducted to identify the domains and items of eSM. Based on our review, we have designed an mHealth app framework that enhances eSM.

This study aimed to assess the practicality and usability of the proposed mHealth app framework designed to enhance eSM through an expert panel review. The framework was reviewed and validated by an expert panel that was specifically formed for this purpose.

The new mHealth app framework will aid developers in creating a practical and user-friendly app to assist patients with epilepsy in effectively managing their condition. Furthermore, it will support HCPs in making informed decisions.

## Material and methods

### Study design and strategy

This study presents the validation process of the proposed mHealth app framework for epilepsy self-management using content validity through experts.

In this study, we developed a mHealth app framework for epilepsy self-management, which was created based on:

1. Initially, a systematic review of all mHealth apps for eSM available on Google Play for Android and the App Store for iOS was conducted to identify the common domains and features of epilepsy self-management. This review was subsequently published [18].

2. Next, a literature review was conducted to identify the domains and items related to eSM.

### Materials

In this study, a mHealth app framework is proposed, encompassing the following domains:

**Seizure Tracking (ST).** The Seizure Tracking domain provides crucial data that help medical professionals assess the medication and medical care received by patients [19].

**Medication Adherence (MA).** Nonadherence remains a significant concern for patients with epilepsy [20], as it can lead to an increase in seizure frequency [21]. In general, patients with epilepsy do not adhere to their medication, which has adverse effects on their condition [22]. Nonadherence to medication, in turn, often leads to poorer treatment outcomes, more frequent consultations and hospitalizations, and increased costs.

**Treatment Management (TM).** Treatment management involves helping patients remember important aspects of their medical care, such as appointments, advice, and suggestions received from HCPs [18], as well as being aware of health care costs [22].

**Health Care Communication (HCC).** The patient-centered paradigm thrives when HCPs and patients engage in effective communication with each other [23]. This not only strengthens their relationship but also enables the delivery of superior care.

The framework includes 21 items related to eSM, which are discussed in Table 1.

### Proposed framework

Table 1.

### Procedure

This study was performed on January 7, 2023 till August, 2023 which consisted of two stages: content validation and panel interview.

**Table 1. Domains and items of the proposed mobile health application framework that enhance epilepsy self-management.**

| Domain | Items |
|---|---|
| Seizure Tracking (ST) | 1. Keeping track of seizures [24] |
| | 2. Keepingtrack of the frequency of seizures |
| | 3. Keepinga record of the types of seizures |
| | 4. Identifying circumstances that could lead to seizures |
| | 5. Conscious about circumstances or items that could trigger seizures |
| | 6. Contacting HCP[a] about side effects |
| Medication Adherence (MA) | 1. Having medication readily available for seizures |
| | 2. Taking seizure medication as recommended even during exceptional events (e.g., holidays, birthdays, vacations) |
| | 3. Taking seizure medication at the same time every day |
| | 4. Using seizure medication as directed by the medical professional |
| Treatment Management (TM) | 1. Being on time for doctor's or clinic visits |
| | 2. Finding techniques to recall the tasks at hand |
| | 3. Getting the seizure medication refilled on time |
| | 4. Completing tests as directed by a medical professional (e.g., blood tests) |
| | 5. Adjusting or changing medication when it causes side effects via a phone call without visiting the doctor (self-developed) |
| | 6. Talking with someone about epilepsy/seizure when the need arises |
| HealthCare Communication (HCC) | 1. Discussing with the HCP about the duration of the epilepsy therapy |
| | 2. Discussing with the HCP about how to use seizure medications |
| | 3. Talking to the HCP about sleeping habits |
| | 4. Sending weekly and monthly reports through WhatsApp to HCP (self-developed) |
| | 5. Talking to the HCP about emotions |

[a]HCP: health care professional.

## Participants

Five experts, including clinicians and pharmacists from academic universities, as well as representatives from the Malaysian Society of Epilepsy (MSE), a patient-based non-government organization (refer to Table 2), were invited to be part of the expert panel. They were contacted through email and provided with a content validation form (see S1 Appendix). The selection of experts was based on their educational and professional backgrounds, as well as their extensive experiences in epilepsy management [25].

**Table 2. The education level, occupation, and work experience of the expert panel members.**

| Expert | Education | Occupation | Work experience (years) |
|---|---|---|---|
| 1 | Doctorate | Doctor (epileptologist) | >15 |
| 2 | Master's | Doctor (MSE[a]) | >15 |
| 3 | Doctorate | Doctor (epileptologist) | >15 |
| 4 | Doctorate | Pharmacist | 10 to 15 |
| 5 | Master's | Pharmacist | >15 |

[a]MSE: The Malaysian Society of Epilepsy.

**Table 3. The degree of relevance.**

| |
| --- |
| 1. The items do not apply to the measurement domain. |
| 2. The item has a tenuous relationship to the measurement domain. |
| 3. The item has a strong connection to the measurement domain. |
| 4. The item is very pertinent to the domain being measured. |

## Validation process

**Ethical statement.** The authors are accountable for all aspects of the work in ensuring that questions related to the accuracy or integrity of any part of the work are appropriately investigated and resolved.

*Content validation.* The experts assessed the practicality and usability of each item in all domains [26]. Content validation by experts involves gathering well-informed opinions from individuals experienced in the subject matter, recognized as knowledgeable authorities, and capable of providing data, proof, opinion, and evaluation [25]. Expert evaluation entails seeking feedback and viewpoints from multiple people on a tool or specific element [27]. Information from experts can be collected individually or as a group, making it either qualitative or quantitative [27]. Content validation can be conducted through face-to-face approaches, such as email, or online methods [28].

Each expert will be requested to evaluate the items of this framework (S2 Appendix) using relevance-related forms 1 to 4, with form 4 being the highest level of relevance (Table 3).

**Panel interview.** Experts were required to provide explanations for their scores on each item, and a final consensus was reached to decide whether to keep or exclude the item.

## Ethics considerations

This study was approved by the University Malaya Medical Ethics Committee (MECID. No. 20181015–6754). Informed consent was taken from all participants before taking part through the email. This study conformed to the provisions of the Declaration of Helsinki (as revised in 2013).

The study was conducted in accordance with the Declaration of Helsinki (as revised in 2013).

## Results

## Content validation by expert

The most common method used to measure content validation is the calculation of the item-level Items Content Validation Index (I-CVI) [29], which is given as follows:

I-CVI formula = Number of experts giving 3 or 4 to relevant items/Number of experts

Based on the I-CVI, items that were scored >0.79 will be considered appropriate, those between 0.70 and 0.79 will require revision, and those <0.70 will be eliminated. The I-CVI calculation results showed that 15/21 items were appropriate. The remaining 6 items were scored less than 0.70, and thus, were eliminated.

Table 4 shows the validation result, following which 6 more items were eliminated.

In the second round, the content analysis tool was used to identify the emergence of any new themes.

The results of the first round indicated the degree of relevance of each item to the domains of the proposed framework. Furthermore, the experts offered new comments and further

**Table 4. The relevance rating of the items by expert and calculation of the I-CVI[a].**

| Domains | Items reflect the domain | E1 | E2 | E3 | E4 | E5 | Relevant | Not relevant | I-CVI | Interpretation |
|---|---|---|---|---|---|---|---|---|---|---|
| Seizure Tracking (ST) | 1. Keeping track of seizures [24] | 4 | 4 | 4 | 4 | 4 | 5 | 0 | 1 | Appropriate |
| | 2. Keeping track of the frequency of seizures | 3 | 3 | 1 | 4 | 3 | 4 | 1 | 0.80 | Appropriate |
| | 3. Keeping a record of the types of seizures | 3 | 2 | 4 | 2 | 3 | 3 | 2 | 0.60 | Eliminate |
| | 4. Identifying circumstances that could lead to seizures | 4 | 4 | 4 | 3 | 4 | 5 | 0 | 1 | Appropriate |
| | 5. Conscious about circumstances or items that could trigger seizures | 4 | 3 | 4 | 4 | 3 | 5 | 0 | 1 | Appropriate |
| | 6. ContactingHCP[b] about side effects | 3 | 2 | 3 | 4 | 4 | 4 | 1 | 0.80 | Appropriate |
| Medication Adherence (MA) | 1. Having medication readily available for seizures | 4 | 4 | 4 | 4 | 4 | 5 | 0 | 1 | Appropriate |
| | 2. Taking seizure medication as recommended even during exceptional events (e.g., holidays, birthdays, vacations) | 4 | 4 | 2 | 4 | 3 | 4 | 1 | 0.80 | Appropriate |
| | 3. Taking seizure medication at the same time every day | 3 | 3 | 2 | 3 | 2 | 3 | 2 | 0.60 | Eliminate |
| | 4. Using seizure medication as directed by the medical professional | 4 | 4 | 4 | 4 | 4 | 5 | 0 | 1 | Appropriate |
| Treatment Management | 1. Being on time for doctor's or clinic visits | 4 | 4 | 4 | 4 | 4 | 5 | 0 | 1 | Appropriate |
| | 2. Finding techniques to recall the tasks at hand | 3 | 4 | 2 | 4 | 1 | 3 | 2 | 0.60 | Eliminate |
| | 3. Getting the seizure medication refilled on time [24] | 4 | 4 | 4 | 4 | 4 | 5 | 0 | 1 | Appropriate |
| | 4. Completing tests as directed by a medical professional (e.g., blood tests) [24] | 4 | 3 | 3 | 2 | 2 | 3 | 2 | 0.60 | Eliminate |
| | 5. Adjusting or changing medication when it causes side effects via a phone call without visiting the doctor (self-developed) | 3 | 4 | 3 | 2 | 3 | 4 | 3 | 0.80 | Appropriate |
| | 6. Talking with someone about epilepsy/seizure when the need arises [24] | 3 | 3 | 3 | 2 | 3 | 4 | 1 | 0.80 | Appropriate |
| Health Care Communication (HCC) | 1. Discussing with the HCP about the duration of the epilepsy therapy [24] | 2 | 3 | 2 | 3 | 2 | 2 | 3 | 0.40 | Eliminate |
| | 2. Discussing with the HCP about how to use seizure medications [24] | 3 | 3 | 3 | 2 | 3 | 4 | 1 | 0.80 | Appropriate |
| | 3. Discussing with the HCP about sleeping habits [24] | 2 | 2 | 2 | 2 | 3 | 1 | 4 | 0.25 | Eliminate |
| | 4. Sending weekly and monthly reports through WhatsApp to HCP (self-developed) | 3 | 3 | 4 | 4 | 4 | 5 | 0 | 1 | Appropriate |
| | 5. Talking to the HCP about emotions [24] | 3 | 4 | 3 | 2 | 3 | 4 | 1 | 0.80 | Appropriate |

[a]I-CVI: Items Content Validation Index.
[b]HCP: health care professional.

information from the second round, which significantly contributed to the improvement of the proposed framework.

The framework was adjusted and improved based on the outcomes of the two rounds.

This study aimed to validate the propose framework that enhances eSM and assesses content validity using expert comments. The input from experts played a crucial role in clarifying, adding, and modifying essential aspects of the framework.

During the first round, the experts were asked to validate the items in each domain and provide scores ranging from 1 to 4, with 4 indicating high relevance to the domain. This round yielded quantitative data, which led to a reduction in the number of items from 21 to 15.

In the second round, the experts were requested to provide comments and recommendations for the framework. As a result, three valuable suggestions and remarks were received from the professionals. After analyzing the data obtained from the experts, we concluded the following:

- Many patients with epilepsy face challenges related to seizures and side effects. One expert highlighted that seizures and side effects are the most common issues reported by patients, often requiring clinical visits.

- Based on input from the interviewed experts, the item "side effect" was suggested to be placed under the Medication Adherence domain. As a result, we transferred this item accordingly to align with their recommendations.

- Several factors can contribute to the occurrence of seizures. One expert emphasized the significance of recognizing situations that might trigger a seizure. This awareness is vital as it alerts patients to potential factors that could lead to a seizure.

- Most patients found it difficult to identify the type of seizure they were experiencing/experienced. Thus, they were not asked to record it.

- Medication adherence is one of the most importance, with the critical aspect being the timing of taking the medication. One expert stressed the significance of regular medication intake.

- Taking the medication daily is considered a critical issue. Therefore, all experts unanimously agreed that daily medication intake is highly important, and the specific time of taking the medication does not matter.

   Within the Treatment Management domain, there are six items, and one of the items that experts extensively discussed was "medication adjustment or change."

- Treatment adjustment is defined as (1) increasing the number of medication classes recommended, (2) changing the usage of at least one ongoing medication class, (3) switching to a medication in a different class, or (4) switching to a different medication within the same class over the course of 12 months during the intervention.

- During the interview, one of the significant points and a common argument among experts was related to medication adjustment. As a result, this item was relocated to the Health Care Communication domain based on their recommendations.

- The adjustment of medication through communication with HCPs or via an mHealth app will enhance eSM and reduce the risk of drug interruptions for patients with epilepsy and other individuals dealing with chronic disorders.

- Making modifications to the pharmacological therapy of well-managed patients with epilepsy who have expressed multiple concerns leads to an improved quality of life [30].

## Discussion and limitations

mHealth apps play a primary and crucial role in epilepsy self-management [31]. We have developed an mHealth app framework designed to assist PWE in enhancing their self-management.

   Our aim was to assist app developers in understanding the essential domains that should be incorporated into apps supporting eSM.

   The results of this study demonstrated that seizure tracking is one of the crucial domains of eSM, aiding health care providers in making informed decisions and facilitating epilepsy management.

   According to the Institute of Medicine (IOM), tracking seizures is a critical component of seizure management [32].

The findings of this study demonstrate that medication adjustment by patients themselves is highly significant. A study conducted during the coronavirus disease 2019 (COVID-19) pandemic found that over 40% of the patients reported that contacting their HCPs for epilepsy management, stress control, and seizure management became "much more difficult" during the epidemic, despite not requiring appointments to visit doctors at that time [33]. PWE follow-up should be frequent but not burdensome for doctors, and a mHealth app can bridge this gap [34].

Medication adjustment or change typically takes place during follow-up appointments. However, under certain circumstances, only sick patients are accepted for appointments with doctors, and follow-up may not be available. This was particularly evident during the COVID-19 pandemic [33].

Drug self-management is a crucial factor in improving patient control and seizure management [35]. In a survey conducted by Xi Liu [36] regarding mHealth apps for epilepsy self-management among PWE, it was found that 66.7% of the sample indicated that managing their seizure through an app would be useful. mHealth apps should not only provide information to enhance the patient's understanding of their condition but also empower them for better self-management [31, 37]. A study by Choi and colleagues [37] revealed that a significant majority of participants struggled to recall how often they missed their medications. Medication reminders were found to be a helpful feature in improving medication adherence [37], which can address the challenges faced by HCPs in ensuring medication adherence.

Sharing emotions with doctors was recommended as a priority. Psychological symptoms are critical issues that patients with epilepsy often encounter. Moreover, when generating and sharing information, data should be treated as "highly confidential" to ensure the privacy of the patient's data and personal information.

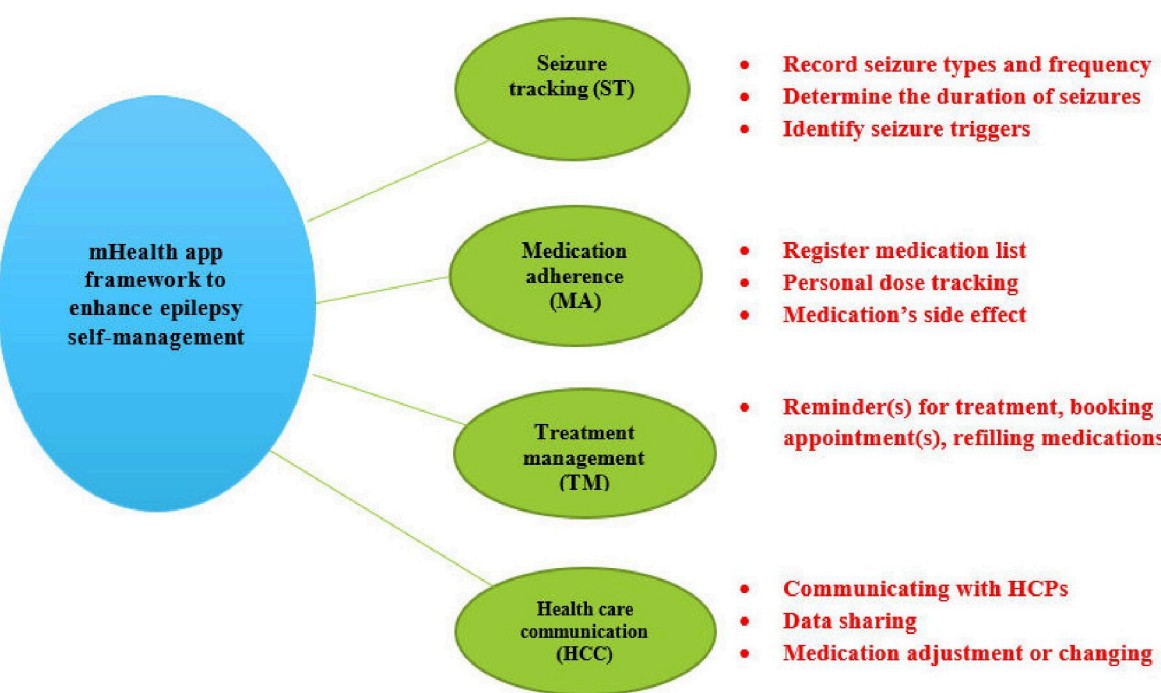

**Fig 1. The developed epilepsy self-management mobile health app framework.** The figure also presents the four domains and their various items (see text and Table 1 for additional details).

This research found that incorporating a calendar feature within the app and setting timely alerts can greatly assist patients in effectively managing their condition.

After incorporating feedback from rounds 1 and 2, the final framework was generated. Fig 1 presents the final eSM framework, which will serve as a guide for mHealth app developers in creating an app that effectively supports patients with epilepsy in enhancing their self-management.

One limitation of this study was the small number of experts, which could potentially impact the study results.

## Conclusions

The primary objective of this study was to validate the proposed mHealth app framework by gathering opinions from experts. The components of the framework were derived from a previous study and existing literature, as well as app-related information. Expert feedback and comments played a crucial role in further enhancing the proposed framework.

Experts' validation proves that Seizure Tracking, Medication Adherence, Treatment Management, and HealthCare Communication are the most important domains that help in enhancing eSM.

Future work should implement the proposed framework and obtain the evaluation from end users.

## Supporting information

**S1 Appendix.**
(DOCX)

**S2 Appendix.**
(DOCX)

## Acknowledgments

We would like to thank all experts who participate in this study for their time and efforts.

## Author Contributions

**Conceptualization:** Mohsen Zaied Alzamanan, Kheng-Seang Lim, Maizatul Akmar Ismail, Norjihan Abdul Ghani.

**Data curation:** Mohsen Zaied Alzamanan, Kheng-Seang Lim.

**Formal analysis:** Mohsen Zaied Alzamanan, Kheng-Seang Lim, Maizatul Akmar Ismail, Norjihan Abdul Ghani.

**Funding acquisition:** Mohsen Zaied Alzamanan, Kheng-Seang Lim.

**Investigation:** Mohsen Zaied Alzamanan, Kheng-Seang Lim, Maizatul Akmar Ismail, Norjihan Abdul Ghani.

**Methodology:** Mohsen Zaied Alzamanan, Kheng-Seang Lim, Maizatul Akmar Ismail, Norjihan Abdul Ghani.

**Project administration:** Mohsen Zaied Alzamanan, Kheng-Seang Lim.

**Resources:** Mohsen Zaied Alzamanan, Kheng-Seang Lim, Maizatul Akmar Ismail, Norjihan Abdul Ghani.

**Software:** Mohsen Zaied Alzamanan, Kheng-Seang Lim, Maizatul Akmar Ismail, Norjihan Abdul Ghani.

**Supervision:** Mohsen Zaied Alzamanan, Kheng-Seang Lim, Maizatul Akmar Ismail, Norjihan Abdul Ghani.

**Validation:** Mohsen Zaied Alzamanan, Kheng-Seang Lim, Maizatul Akmar Ismail, Norjihan Abdul Ghani.

**Visualization:** Mohsen Zaied Alzamanan, Kheng-Seang Lim, Maizatul Akmar Ismail, Norjihan Abdul Ghani.

**Writing – original draft:** Mohsen Zaied Alzamanan, Kheng-Seang Lim, Maizatul Akmar Ismail, Norjihan Abdul Ghani.

**Writing – review & editing:** Mohsen Zaied Alzamanan, Kheng-Seang Lim, Maizatul Akmar Ismail, Norjihan Abdul Ghani.

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
