## [Decision Letter · Decision Letter 0]

4 Mar 2024

PONE-D-23-41088Development of an Epilepsy Self-Management Mobile Health App FrameworkPLOS ONE

Dear Dr. Alzamanan,

Thank you for submitting your manuscript to PLOS ONE. After careful consideration, we feel that it has merit but does not fully meet PLOS ONE’s publication criteria as it currently stands. Therefore, we invite you to submit a revised version of the manuscript that addresses the points raised during the review process.

This manuscript needs further substantial work; both the reviewers have raised serious concerns about the framework and the statistical part, as well as the references, to name a few. You are advised to carefully make changes suggested by the experts.

We look forward to receiving your revised manuscript.

Kind regards,

Muhammad Farooq Umer, PhD Epidemiology and Health Statistics

Academic Editor

PLOS ONE

Reviewers' comments:

Reviewer's Responses to Questions

**Comments to the Author**

1. Is the manuscript technically sound, and do the data support the conclusions?

Reviewer #1: Yes

Reviewer #2: Partly

2. Has the statistical analysis been performed appropriately and rigorously? 

Reviewer #1: I Don't Know

Reviewer #2: N/A

3. Have the authors made all data underlying the findings in their manuscript fully available?

Reviewer #1: Yes

Reviewer #2: Yes

4. Is the manuscript presented in an intelligible fashion and written in standard English?

Reviewer #1: No

Reviewer #2: Yes

5. Review Comments to the Author

Reviewer #1: This study aimed to assess the practicality and usability of the proposed mHealth app framework designed to enhance Epilepsy Self-Management Mobile Health App through an expert panel review. Thank you for the opportunity reviewing this article. I may require some comments on the following issues.

Abstract:

The findings section is long, in this section only mention the main results.

Check keywords with mesh to make sure they are standard.

Introduction:

The introduction section should be made stronger.

In the second or third paragraph of this section, mention the importance of m-health in the management of chronic conditions. For this purpose, use the following related articles.

https://pubmed.ncbi.nlm.nih.gov/29726434/

https://pubmed.ncbi.nlm.nih.gov/28508813/

https://pubmed.ncbi.nlm.nih.gov/29726430/

Methods:

the following structure should be observed for the method part:

1. Study Design and Search Strategy

2. Data sources

3. Study selection

4. Data extraction

5. Quality and risk of bias assessment

Discussion:

There is considerable gap in discussion section relating to discuss about your findings using related studies. So, use the following articles this section as well.

https://www.sciencedirect.com/science/article/abs/pii/S2211883720300642

https://pubmed.ncbi.nlm.nih.gov/33227362/

https://pubmed.ncbi.nlm.nih.gov/34723447/

Please include limitations section within the discussion section (preferably last paragraph).

Reviewer #2: Developing an Epilepsy Self-Management Mobile Health App Framework is worthy work on the subject; however, the manuscripts lack crucial and fundamental concepts.

1- I'm concerned that the authors overlooked any theories considered when developing the Epilepsy Self-Management Mobile Health App Framework. The authors should address whether any theories are employed to develop the mobile health app framework.

2- Although the authors mentioned that The framework was created following a systematic review of all mHealth apps available on Google Play for Android and App Store for iOS, the manuscript did not address the theoretical framework, which is a fundamental baseline for developing a framework.

3- I could not see the specific processes in the Development of the Development of Epilepsy Self-Management Mobile Health App Framework. The authors mentioned the methodology but not in a sequence.

4- The manuscript lacks a comprehensive conceptual framework for developing the Epilepsy Self-Management Mobile Health App

5- References are not in line.

6. PLOS authors have the option to publish the peer review history of their article (what does this mean?). If published, this will include your full peer review and any attached files.

Reviewer #1: **Yes: **Esmaeil Mehraeen

Reviewer #2: No

---

## [Author Response · Author response to Decision Letter 0]

23 Mar 2024

All the comments were fixed and could you check our respond to the reviewer's comments which has attached with documents.

---

## [Decision Letter · Decision Letter 1]

15 Apr 2024

Development of an Epilepsy Self-Management Mobile Health App Framework

PONE-D-23-41088R1

Dear Dr. Alzamanan,

We’re pleased to inform you that your manuscript has been judged scientifically suitable for publication and will be formally accepted for publication once it meets all outstanding technical requirements.

Kind regards,

Muhammad Farooq Umer, PhD Epidemiology and Health Statistics

Academic Editor

PLOS ONE

Additional Editor Comments (optional):

Reviewers' comments:

Reviewer's Responses to Questions

**Comments to the Author**

1. If the authors have adequately addressed your comments raised in a previous round of review and you feel that this manuscript is now acceptable for publication, you may indicate that here to bypass the “Comments to the Author” section, enter your conflict of interest statement in the “Confidential to Editor” section, and submit your "Accept" recommendation.

Reviewer #1: All comments have been addressed

2. Is the manuscript technically sound, and do the data support the conclusions?

Reviewer #1: Yes

3. Has the statistical analysis been performed appropriately and rigorously? 

Reviewer #1: N/A

4. Have the authors made all data underlying the findings in their manuscript fully available?

Reviewer #1: Yes

5. Is the manuscript presented in an intelligible fashion and written in standard English?

Reviewer #1: Yes

6. Review Comments to the Author

Reviewer #1: Yes, you can use this reference:

Mehraeen E, Safdari R, Seyedalinaghi SA, Mohammadzadeh

N, Arji G. Identifying and validating requirements of a

mobile-based self-management system for people living with

HIV. Stud Health Technol Inform 2018; 248: 140-147.

7. PLOS authors have the option to publish the peer review history of their article (what does this mean?). If published, this will include your full peer review and any attached files.

Reviewer #1: **Yes: **Esmaeil Mehraeen
